# Mystic on a Tilting Stage: Julian of Norwich's Performance of English Visionary Devotion

Elizabeth F. Perry

Department of English, Texas A&M University, College Station, TX 77840, USA; efperry992@tamu.edu

**Abstract:** Julian of Norwich's performance within her longer *Revelations of Divine Love* involves layers of authorizing and devotional steps that frame it as a gift for her community. She presents herself not as an author, but as a revelator, in step with John's acts of unveiling his visions and dialogue with the divine in the Biblical Revelations. Examining Julian's act of presenting her visions in writing demonstrates how her daring yet insistently orthodox visions handle issues of spiritual authority and individual faith made urgent by the rise of Lollardy. My work with Julian's Revelations is the foundation for a wider argument about the interchange between vernacular mysticism and public devotion through their use of affective piety and the performance of spiritual dialogue. In this article, I examine Julian of Norwich's *Revelation of Divine Love* to determine how it works as contemplative drama. I also look at *The Mirror of the Blessed Life of Christ* and *The Cloud of Unknowing* to set up Julian's performance of contemplative devotion and the potential pitfalls of a pious English readership. Julian's revelations demonstrate where interior contemplation is transformed into collective acts of devotion.

**Keywords:** mysticism; medieval English; Christian contemplative practice; apocalypse; devotional performance; women's authorship

## 1. Introduction

When Julian of Norwich closes her *A Revelation of Love* with the observation that Christ's revelation through her "is not yet performed, as to my sight" (Julian of Norwich 2006, edited by Watson and Jenkins, LT Chapter 86, ll. 1–2), she opens her visionary experience to a practice of common devotion.[1] She imagines herself and her Christian readership growing into revelation through careful steps of "seeing" and performance that leads deeper into the love of God. Julian imagines her vision as the beginning of further revelation. Christ has created a glimpse of truth for her readers, "which knowing he wille geve us grace to love him and cleve to him" (86, l. 7). In many ways, Julian is practising a common Christian tradition that informs adherents how their sight, disciplined through prayer, becomes wisdom. Imagined glimpses of Christ leads to knowledge, which further leads to a desire to grow closer to God through God's love. Julian's vision of a continued performance of Christ's love moves her revelation from her exclusive mystic experience into a contemplative, pious performance accessible to her readers.

This practice of devout imagination, which Nicholas Love best detailed in his English treatment of the *Speculum Vita Christi*, combines scripture, exhortation, and colourful narration. Love formats his book, Richard Beadle argues, in order to bring "simple souls closer to God and stimulated a desire for heaven" (Beadle 1997, p. 11). The devout soul is active in imaginative "seeing" of the spiritual truths that are contained in the Gospel. Nicholas Love urges his readers to be present fully with the life of Christ in their imagination through prayer and reflection on examples of Christ's love and virtue in the Gospels. Julian of Norwich's long-text treatment of her visions represents the most remarkable playing out of the devotion illustrated in Love's work. Julian's work is not an interior mediation but a

creation of a visible, dramatic spiritual space within and all around her body and soul that she calls all her readers to inhabit.

The instructions that English examples of imaginative devotion give to souls seeking God reveal their essential performative nature. The devotional steps outlined in the English pseudo-Bonaventure translated by Nicholas Love, *The Mirror of the Blessed Life of Jesus Christ*, allow the simplest vernacular devotees to place themselves into the drama of the Incarnation of Christ. The process begins with meditating on the love of God as revealed through his presence. The devout then repeat their meditations through action and the creation of spiritual spectacle. Participation creates an understanding of the heights and depths of God's plan of love as revealed through the life of Jesus. Nicholas Love crafted his translation and expansion of Bonaventure "in Englyshe to lewde men & women & hem that bene of symple undirstondyng" (Sargent 2005, *The Mirror of the Blessed Life of Jesus Christ*, p. 10). *Mirror* focuses Love's readers attention on scenic detail, urging them to ruminate on well-known prayers and stories from the Bible by imagining themselves in the drama. Love uses biblical narratives and the cycle of the church year to help his readers ponder "þe monhede of cyrste" (Sargent 2005, *The Mirror of the Blessed Life of Jesus Christ*, p. 10). Nicholas Love organises his *Mirror of the Blessed Life of Jesus Christ* around the days of the week. He divides each day to reflect on the bible stories he uses to draw his reader's attention to each holy mystery. There is no distinction between when Love quotes from scripture and when he invents colourful images that would allow his readers to imagine themselves in the action of the Gospel.

Julian of Norwich invites her readers to participate in her visions by straining to capture the ineffable nature of Christ's divine love. Like Nicholas Love, she turns back to the scripture. But the depth of revelations leads her not into controlled prayerful meditations as in Love, but a descriptive space where Julian can "garrulously talk her way into the silence of God's mystery" (Turner 2011, p. 25). Rather than using the text merely as a germination sight for imagery and prayer, Julian employs her study of the scripture to guide her readers through challenging moments in her mystic experience. She uses concepts of presence, most notably of sight, to tie her visions to the common act of Christian mediation. As Christopher Abbot explains, Julian equates the presence of God to her ability to see him clearly, describing how "faith is understood not only in revealing God to Julian but a divine perspective, revealing Julian to herself" (Abbot 1999, p. 15). This article argues that Julian, presented in the clarity provided by her visions, is of a revelatory performer. She acknowledges her Christian community as co-spectators that can move within the function of Christ's love in practice. By focusing on Julian as a performer of mystic devotion, I draw attention to how the medieval English Christian community performed their spiritual devotion. Julian follows this pattern, which begins with imagery and moves into embodied and visceral practice. Julian's Bonaventuran desire to be present with the suffering of Christ leads her to create a performance space that deserves exploration.

The style of Julian of Norwich's presentation is inherently artistic and related to performance forms. Her treatment of her visions is so tuned to a cyclical and expressive style that it prompted Denys Turner to relate her work to the sonata form in music. The visions and their meanings are a space where she "is to be found testing. . . and releasing the potentialities of the basic thematic materials. . . or transforming the tiniest melodic materials features of each shewing into major thematic developments" (Turner 2011, p. xi). The idea of Julian playing or performing her visions seems inherent to understanding how she brings her readers close to revealed wisdom. My work focuses on the embodied nature of Julian's work. The performative language and space her visions create allow for an intimacy between Christ's revelations and his people. This brings Julian close to the dramatic language of the Middle English civic cycles, which saw the biblical stories laid over the lived experiences of the cities and their inhabitants. Juliana Dresvina rejects the importance of the cycle plays to the visual language of *Revelations of Divine Love*. She cites the busy and bombastic nature of the medieval playing space compared with the visions given to Julian, which are "never crowded with figures; they are focused on one maximum of two objects"

(Dresvina 2019, p. 6). The focus Julian's visions place on their subjects is undeniable and vital to the theology-through-devotion model she practises. However, Julian's positioning conjures a different connection to medieval religious drama—the primary role of the spectators. In her longing focus on her objects of devotion, Julian of Norwich models a proper mode of spectatorship, especially for viewing the passion. She stands in a crowd of the community of believers for whom her work is intended and participates alongside them in the salvific drama of the cross. Julian of Norwich's work through her visions ties her closely with the language of devotional drama.

## 2. Julian's *Revelation* as an Endless Practice

Julian's visions represent vivid participation in the tradition of imaginative devotional performance both for Julian and her readers. Devotion as a performance is not unique to Nicholas Love. In this text, Julian practices a style of contemplation innate to a wide array of texts, images, and performances from this period. She hopes to guide those who participate with her to reading into her longing for God. Kirsty Clarke notices how both Julian and the austere spiralling language of the anonymous *Cloud of Unknowing* "contain within them the potential to recreate the mystical experience in their readers" (Clarke 2019, p. 106). The texts' participatory potential continues and expands long past the earthly life of a single contemplative author. Notably, the long text of Julian's work survived through the devotional translating and practice of the English Benedictine nuns in Cambrai and Paris before it was translated into the 1670 Cressy edition for the private devotional practice of Lady Mary Blount.

Jennifer Summit's exploration of the temporal transcendence of Julian's survival argues for a contemplative performance history of Julian's visions. Summit's reading reveals how the contemplative performance of Julian's work preserves a "shared experience of beholding" (Summit 2009, p. 32). Cressey's translation foregrounds Lady Blount's private devotion, tying her to Julian in his dedication by introducing the fourteenth-century mystic as "a Person of your own Sex, who lived about Three Hundred years since". He stresses to Lady Blout how Julian wrote her visions "for You, [emphasis in the original] and for such Readers as your self" and hopes Blount and Julian will share a devotional space akin to "conversation" (Julian of Norwich 1670, published by Cressey, fol. A2r). Cressey presents Julian as both a conveyor of spiritual wisdom and a perspective through which a pious soul can better position themselves into the drama of Christ's revelation. The book invites Julian's readers to share in both her visions and the growing longing their revealed knowledge gives them for the presence of God.

Julian of Norwich reminds her readers, her "even-Christen"[2], that she began her journey that culminated in Christ's revelations with the same desire that Nicholas Love urges for the readers of his *Mirror*. Julian longs for a full understanding of Christ's passion. She strives for participation, placing her body and her spiritual sight into action in a search for God. Julian's desire to "have more knowinge of the bodily paines of our saviour, and the compassion of our lady, and all his true lovers that were living that time and saw his paines" (Chapter 2, ll. 10–12) predates Love's *Mirror* by decades. Yet it encapsulates the full drama Love hopes to inspire in his readers. Julian longs to be within the pain of Christ through her entire body and discovers for her readers the fullest possible portrait of the love of Christ through her experience. Julian's *Revelation of Divine of Love* uses Julian's visions to set up a series of contemplative performances to guide her readers into a close, imaginative experience of Christ's love. She moves through her visions already bound into Christ's love but strives more and more towards the contemplative unity that passes the performance of her visions and will only be perfect after her death. As Kirsty Clarke illustrates, Julian is in motion in her striving until the point where she has "reached the stage where she can leave the image behind and enter into contemplative prayer; a prayer without words" (Clarke 2019, p. 103). Rather than a guidebook for her fellow lay readers, the *Revelations* present Julian as active and acting out a revelatory performance. Her meditation and

subsequent reflection on her vision of the divine creates a stage space where her readership is led into a full understanding of the love of God.

Julian of Norwich presents her work as a divine revelation of the core of Christian faith. Through her visions, Julian receives knowledge about the nature of Christ, his suffering, and his divine grace. However, God reveals himself to Julian through more than her ghostly sightings. According to Julian, Christ graces her intellect with God's wisdom. This makes her contemplations divinely inspired. Rumination on her visions—guided by meditation and scripture—gives her deeper knowledge of God's love. The meeting of her desire through prayer with Christ's revelatory wisdom allows her to communicate with the Christian community. What her long text presents is the labour of many years' transformation of "bodily sight" into an understanding of the substance of God's love. Rather than positioning herself as a theologian or even against false heretics, like many of her contemporaries, including Nicholas Love, Julian presents herself as a working body and spirit in motion, transmitting back the simple truth of God's love. Jessica Scott presents a delightful, stage-like image of Julian in motion through her discernment. She imagines Julian "repeatedly pivoting, perhaps even stumbling around what she sees. She returns and re-enters the scene, understanding from which will emerge only as she moves through time" (Scott 2022, p. 488). Julian's writing manifests God's charity through practice. She imparts what has been revealed to her through prayer and study to benefit the faithful. *A Revelation of Love* crafts a complex revelatory exercise: God imparts the nature of love to Julian of Norwich first through images and then through her efforts to reveal love to the Christian community.

Julian's long period of contemplation on her visions allowed her to reveal the knowledge she has gained in a way that draws her readers into an immersive staging of her personal revelation. The image of the Saviour's skin changing colour as it dries while red pellets of blood flow out of him like rainwater frames Julian's initial vision within the intense affective piety common to the fourteenth century. Julian writes guided meditations on the symbolic imagery she receives. She accompanies her vision of the created order as a hazelnut with an insight. The vision contains "three properties: the first is that God made it, the secund is that God loveth it, and the thirde is that God kepeth it" (Chapter 5, ll.14–15). This insight leads Julian to a deeper understanding of the vastness of God's concern for human smallness. Julian participates in these visions in her later work in a way that incorporates an understanding of how communal performance impacts the drama of salvation. Sarah Beckwith references Julian of Norwich as an example of how "theater [sic] and church are deeply communal enterprises. . . neither can be understood except as a performance of that community" (Beckwith 2001, p. xvi). Beckwith here indicates how Julian's creation of meditative images demands a community to perform what has been revealed through her. Reading Julian's *Revelations* transforms the reader into a participant. The visions guide the reading participant to a message about God's love. Julian explores the images revealed to her in a way that makes their meaning as full and evident as possible to her Christian community.

God grants knowledge to Julian through what she sees as the grace to interpret and gain understanding about divine nature from her visions. This chain of knowledge does not eliminate Julian as the writer of these texts. By presenting herself initially as "a simple creature unlettered", Julian demonstrates that she was able to make sense of her visions, another representation of charity that God has chosen to impart through her writing (Chapter 2, l. 1). Importantly, she is not a teacher. The work she has done with her visions is not one of creation. She stresses in her work that she is not uncovering anything that is a stumbling block to her reader's understanding. Everything Julian knows, she does so because God first desires her to see it.[3]

God acts within Julian, moving lastly to position her to have a full view of herself. God opens her "ghostely eye and shewde me my soule in the middes of my harte" when she is faced with the threat of doubt and fear of her future sins. Without the knowledge imparted to her, Julian is a stranger even to her own heart (Chapter 68, ll. 1–2). Since God, in Julian's

estimation, represents "endlesse souereyne truth, endlesse souereyne wisdom endlesses souereyne love unmade", the wisdom displayed in her theology demonstrates a loving gift from God (Chapter 44, ll. 9–10). She, the eager student, first learns "to chese Jhesu for my heven", and then to decipher what his word says to her community (Chapter 19, l. 12). Through this instruction, Julian is meant to show her readers how they are meant to receive the work she has done, uncovering God's words on their behalf.

Structurally, Julian's designation of her theological exercises as a revelation supports her right to receive and interpret holy visions. Anna Lewis argues that Julian emphasises the received nature of her knowledge. If her conclusions are "illuminated, led, and inspired by God, the phrase 'as to my understanding' can be seen less as a qualification than as a statement carrying some authority" (Lewis 2009, p. 82). Chapter LX, which contains Julian's exploration of the motherhood of Christ, uses the device of Julian being shown this truth "as I understonde in the mening of oure lord" (Chapter 60, l. 2). However, Julian also employs a careful and detailed exploration into the scripture that reinforces her conceit. Julian seems excited, not defensive, as she explains, "oure very moder Jhesu, he alone bereth us to joye and to endlesse leving—blessed mot be he!" (60, ll. 15–16). As Denise Baker explains, Julian's work ventures beyond visions of the manhood of Christ into territory daunting to any theologian. What Julian interprets as divine inspiration allows her to ascend to "the point where meditation gives way to contemplation and visionary experience to mysticism" (Baker 1994, p. 61). Julian constructs her contemplation to include instruction in how her visions are loosened and unpacked as comfort for those she has been stirred to write.

Julian wrote her *Revelation* in order to communicate the truth she received to her community of readers. Julian sets up her revelations as a journey she undergoes towards understanding what she has witnessed. She begins this journey by actively seeking to know more of Christ's suffering. Julian recounts, "I had sundelee feeling in the passion of Christ but yet desired to have more by the Grace of God" (Chapter 2, ll. 5–7). Julian embarks on a discovery of all the meaning contained in Christ's definitive act of divine love and what it implies for sinners. This is not an easy task for her, even with divine inspiration. As Barry Windereat explains, for Julian, "there was only ever one subject. . .understood cumulatively over time in response to the various unclarities, problems and challenges" (Windeatt 2004, p. 72). Julian is upfront with questions she has about the nature of mercy and free will, especially concerning God's relationship to human sin. Her question—"why, by the grete forseeing wisdom of God, the beginning of sinne was not letted" (Chapter 27, l. 5)—remains a common concern among Christian thinkers. Julian deftly turns the question over to Jesus, who "enformed me alle that me neded" and, by so doing, informs all of Julian's readership (27, l. 9). Denise Baker further sees an outward focus in Julian's *Revelation* in the way she structures her showings amongst her meditations on their meaning. Every facet of her visions combines to reinforce Julian's assertion that God's grace will make all things right and "manifest the essential unity of her vision" (Baker 1994, p. 157). Julian reveals God's benevolent mission to her readers in an accessible manner.

Love is the ultimate reason Julian gives for transforming her visions into her *A Revelation of Love*. God's love, referred to in the medieval English vernacular as "charitie", reveals itself through works of charity within the Christian community.[4] Julian communicates God's unending charity first by the act of presenting truth and comfort to the Christian community and second by expounding on the nature of this comfort in her book. Understanding, she claims, came through intense struggle as she "desyerde oftentimes to witte what was oure lords mening" and for fifteen years she was given the same answer: "love" until she "sawe full sekerly" (Chapter 86). Her great act of charity in writing is to help her readers avoid the same frustration. Julian's revelation leans on the authority of the church and the church doctors to fill in where she had been left with an incomplete sight, reminding her readers to stand in trust and dread of God. To provide commentary would be to assume authority, so instead, Julian channels what she has learned back through its

original source. In Chapter XLII, Julian leads her readers on an in-depth exploration of how best to pray. Although the explanation is clearly Julian's work, she begins the work restating that "this is oure lords wille" (Chapter 42, l. 11). She then uses this authority to remind her readers that the true power of prayer comes directly from God.

Julian of Norwich presents herself in her *Revelations of Divine Love* as participating in her contemplation as an interpreter for contemplative readers. Anna Lewis argues that Julian's appeal to the Christian body indicates the work is meant to be interpreted by the entire community of believers. This would make Julian's hoped-for act of charity in presenting this book a model of devotion for "a Christian body made up of people who think and feel like she does" (Lewis 2009, p. 86). While not many Christian readers could hope to approach the contemplation needed to create a work like *A Revelation of Love*, Julian presents her offering in such a way that communicates abstract, difficult concepts to Christ's followers. She acts out her visions in a way that carries the performative tradition of English devotion into a shared experience.

### 3. The Mystic in Performance

Julian's A *Revelation of Love* presents Julian acting as both a mystic and a Christian revelator. Though, traditionally, presentations of heavenly visions were meant only to relay the voice of God, Julian's serves as far more than just a mouthpiece for her showings. Julian burns with a desire to move her feeling of love for Christ in his passion into "bodely sight wherein I might have more knowinge of the bodily paines of our savior" (Chapter 1, ll.10–11). A full understanding of the love of God, as presented through the suffering body of Christ, requires Julian's full physical body to be participating alongside those who lived alongside and loved Christ. Julian's mission is to be an active presence within the performance of God's revelatory love to the faithful. Julian is acknowledging a characteristic innate to affective piety practices which, as Julia Dresvina describes, are "grounded in everyday materiality, the inevitability of the embodied-ness, embedded-ness, of the mind, of the word made flesh" (Dresvina 2019, p. 15). She asks through the grace of God that her visions move her full mind-body to where she might "be one of them and have suffered with them" (1, ll. 12–13). God grants her desire, moving Julian to present a realised examination of the love of God opened to the faithful. She presents what has been revealed to her as a message to the body as a whole. She transforms the "I" of her visionary experience into a comforting "we" that is meant for all participating in the interplay of God's love for the body of the church. This is how Julian can present her longer work as merely "begonne by Goddes gifte and grace, but is not yet performed, as to my sight" (Chapter 86, ll. 1–2). What God has presented to Julian's bodily sight are visions revealing the nature of Christ's suffering, the formlessness of sin, and the overarching power of love. Through her further study, God has revealed how God will make all things well. Julian's revelation concerns both the whole body of the church and the eternal design outside of time.

An obvious literary predecessor to Julian's participatory revelation style resides in the apocalyptic visions presented by John in the Christian scripture. In his *Revelations of Jesus Christ*, John claims to be a witness to visions presented to him by God "to show his servants what must soon take place" (Revelation 1:1). John deliberately establishes himself as a reporter who records the visions given to him dutifully—sometimes taking direct instructions on what he is to tell the people: "I heard a voice from heaven say, 'Write. . .'" (Revelation 14:13). John acts on his obligation. He announces himself as the John who has known the person of Jesus and introduces his vision as provided by God for all who follow him. John illustrates how God, not he, expects the visions to be performed by the readers, stating that "Blessed is the one who reads aloud the words of this prophecy, and blessed are those who hear and who keep what is written in it" (Revelation 1:3). John sets up an expectation of how the vision should sit in the mouth, ears, and heart of his readers.

The careful use of specific words is critical when contained in a vision that presents Christ and his program of salvation. In *Revelations*, John warns about how they must all be kept together lest "God will take away his share in the tree of life and in the holy city, which

are described in his book" (Revelation 22:18). This concern surrounding mystic language and the Christian practice of contemplating the divine carries into fourteenth-century English vernacular writing. The challenge appears in both *Revelations of Divine Love* and *The Cloud of Unknowing* as "how embodied finite human beings can understand, experience, and communicate about an infinite, immaterial reality" (Harkaway-Krieger 2021, p. 70). Notably, the anonymous author of *The Cloud of Unknowing* prays at the beginning of his mystic instruction manual for the message contained to be performed correctly. He worries about the use of his specific words, cautioning his readers that if:

> "any soche schal rede it, write it, or speke it, or elles here it be red or spokin, that thou charge hem, as I do thee, for to take hem tyme to rede it, speke it, write it, or here it, al over" (TEAMS Middle English Text Series 1997, *The Cloud of Unknowing*, ll. 21–23).

The author of *The Cloud of Unknowing* attempts to temper the danger of partial or misreading by restricting his readership to those dedicated to contemplation. The concern John expresses about the substance or even the practice of performing the substance of mystic visions continues into the English vernacular in the decades before Julian of Norwich experienced her visions.

Julian of Norwich and John, author of the Apocalypse, take on the daunting task of demonstrating the proper way the whole body of believers should participate in a full glimpse of the divine. John places himself both within and outside what has been revealed to him. John sets up his relationship to his revelation in his opening address, stating how these are visions what "God gave him" to perform a message to the churches he is addressing (Revelation 1:1). After John has reported back the message to each church in his address, he calls them to participate as they read in what he has already experienced: "After this I looked and Behold" (Revelation 4:1). Like Julian, John sets up a dramatic interplay that his readers are meant to emulate. John is physically present for all the actions performed around him. John's visions move him to bodily action. He performs tasks and, like Julian, finds himself pulled into visceral reactions to God's revelation. John is made to eat one of the scrolls and finds his "stomach made bitter" (Revelation 10:10). He is compelled to share the gift of his visions as he is transformed into a vessel of its contents. Julian's earthly relationship with her visions also demonstrates how a contemplative performance inspired by her visions should manifest. Julian is so moved by the nothingness of sin and the overwhelming goodness of God that she bursts out laughing. She reports those physically in the room with her, though presumably not participating in her vision, laughed with her "and ther laughing was a liking to me" (Chapter 13, l. 20). The shared act is the first time within the narrative of her visions Julian expresses her desire to share what she's experiencing: "I wolde that alle my evencristen had seen as I saw" (13, l. 22). Julian's revelation mirrors John's in its purpose and how she is placed within her visions.

The sin inherent to all humanity makes the process of transmitting mystic language more challenging. Julian, troubled in the knowledge that she will sin and transfixed on the image of the cross because "all that was beseid... was ogly and ferful to me, as if had ben mekille occupied with fiends", also follows John in her acknowledgement of her own frailty (Chapter 3, ll. 26–27). In both works, the revelation centres a body that the reader is reminded is distinctly human in its capacities. We see John as fallible even as he transcribes out the perfect revelation, reminding his readers he is within the same faith body he reports back to. He pulls away from the vision of the river of life to demonstrate his presence and his fallibility within the substance of the narrative:

> "I John am the one who heard and saw these things. And when I heard and saw them, I fell down to worship at the feet of the angel who showed them to me, but he said to me, "You must not do that! I am a fellow servant with you and your brothers..." (Revelation 22:8–9)

John's visions pull him about, and angels instruct him on how he, and by extension, the body of the faithful, is meant to respond to the teaching being placed in front of him. John is somewhat unstuck in time throughout his revelation as he acts as a mediator between them and his readers. Julian presents an examination of each of her visions through exploring different aspects of scripture. What God "shews" Julian in her text is not only her initial visions, but the strength to interpret these visions correctly. Alongside her image of the gory, dying Christ, Julian asserts she was presented with a "lesson of love" performed through Julian's revelatory examination (Chapter 6, l. 54). She speaks with confidence that her even-Christen "shall see" (6, ll. 54–55). Julian imparts the sight of Christ's love provided to her through steps of physical performance and love to her readers. She transforms her mystic experience into a sight accessible to her readers and provides a backbone to all the theological wisdom she will discover through her following meditation.

Julian's performance in the model created by John continues in the time she spends in contemplation after her mystic experience. This speaks to the scale in which Julian envisions God's message to humanity to play out in the person of Christ. Any conversation revolving around a vision or enactment of Christ's passion in Julian's period must include the resurrection into God's salvific plan as John reveals it. Sarah Beckwith's brief treatment of Julian's performance in her exploration of the York Passion cycle emphasises how the medieval community of believers are committed to understanding performances of Christ's passion as a "perpetually and always present enactment" (Beckwith 2001, p. 88). Julian's anxiety about the state of sinful souls is tied further to the ongoing conflict within her immediate community in Norwich surrounding Biblical representation. For example, Norwich Cathedral was redesigned through the fourteenth and fifteenth centuries to amplify and depict scenes of the Apocalypse. The new architecture incorporated storytelling where "bosses carry the story from the first twelve chapters of Revelation: the opening of the seven seals, the appearance of the four horsemen and the sounding of the six trumpets, until the 'War in Heaven'" (Sekules 2006, p. 294). Julian's immediate community is one devoted to enacting and spectating reminders about their place within the ongoing drama of God's salvific program.

Julian enacts the perpetuity of God's love as embodied in Jesus' suffering and resurrection as her returning concern over the pain of sin when faced by the infallible love of God. Julian constantly keeps passion in mind as she looks forward to salvific, divine time. This way, she can speak in both the present and in ages to come that "the love wherein he made us was in him fro without beginning, in which love we have oure beginning" (Chapter 86, ll. 21–20). Her body remains present as she moves away from the images of her visions through to understanding brought out of her prayers, tying her practice closely to the process of discarding imagery, which the author of *The Cloud of Unknowing* guides his aspiring contemplative through in his prayers. The devastating simplicity of what Julian presents in *Revelations of Divine Love* echoes the spiritual practice the *Cloud* author encourages for those struggling to keep focused on their uninterrupted longing for God. He tells his readers to:

> "take thee bot a litil worde of o silable; for so it is betir then of two, for ever the schorter it is, the betir it acordeth with the werk of the spirite. And soche a worde is this worde God or this worde love" as they linger in the darkness (TEAMS Middle English Text Series 1997, *The Cloud of Unknowing*, ll. 500–502).

Through Julian's desire to feel Christ's passion, she reconfigures apocalyptic visions around the love of God into a reassurance about a beginning and future defined in love.

Julian waits to participate in a culmination of all that has been revealed to her. Her vision is a preface to seeing it performed through the devotional practice of Julian's readers. As John has been commanded by the voice from heaven to reveal Christ's salvation, Julian is driven in her contemplation to find appropriate words for what she has beheld. God gives Julian her answer: after long trials of discernment, Julian knows that her work is, and will continue to be, an exercise in the love of Christ. She tells her readers how: "in love he hath done all his werkes and in love he hath made alle things profitable to

us" (Chapter 86, ll. 18–20). The Christ that Julian encounters in her visions and in her subsequent meditations is active in a continuing work of salvation. Julian longs for the completion of her visions, a performance of what Christ has begun revealing through Julian in her book. She lives in the hope that Christ will make what has been revealed to her profitable for her community. Julian sees herself after decades within these visions as still at work, revealing the substance contained in the love of God. The simplicity of the message is undercut by God's urging to Julian that "thou shall wit more in the same. But thou shalt never wit therein withouten ende" (86, ll. 15–16). Julian must sustain herself in the love of God to learn more of it, but the knowledge she is promised is more on God's love, and yet still more is forthcoming. Julian sees herself in God's program of love for his people through her writing and in her anticipation of sharing it with her community. She exists within her visions not as a recorder of sights but as an active, mystical presence that embodies the revelation of God's salvation.

## 4. The Body and Spirit of "Sight"

The second body present within Julian's *A Revelation of Love* is the accessible and visceral body of the suffering Christ. Julian directs her revelations through the person of Jesus, beginning with her desire to understand his passion. It is a brilliant act of intimacy that circumvents the author of *The Cloud of Unknowing's* concern that a contemplative soul's inward search for God could "provoke all sorts of psychologically introspective contortion" and end up focused merely on their own fallible inner workings (Turner 2011, p. 200). Julian's performance of inwardness displaces the subject into the dying form of Christ. The oddest moments in her viewing come as Julian is given a view *inside* the material substance of God made accessible by the tortures of the cross. The first inner viewing Julian reveals comes as a view into the wounded side of Christ. The opening made by the spear at Calvary opens in Julian's eye into "a fair, delectable place and large inow for alle mankind that shalle be saved to rest in pees and love" (Chapter 24, ll. 3–4). As tempting as it is to move immediately into interpreting the vision as a symbolic description of how Christ's sacrifice forms salvation, Julian pauses her readers at the physical reality of looking into the body of her saviour. She views Christ as both a penetrable body and a resting place. Julian then dives deeper into the side Christ opens to her. Her eyes move through his opened body up to "his blissed hart even cloven in two" (24, l. 7). As she later describes, God has revealed to her "prevites which himselfe shewed openly" allowing her a glimpse inside the body of his love for the world (24, ll. 3–4). Through this visceral image of the broken, loving body of Christ, he guides Julian, through her "understanding, in part, the blessed gohede, as farforth as he wolde that time" (24, l. 8). Julian's eyes perform her moment through the body of Jesus to his godhead in a manner, not unlike the guidance offered in a manual espousing what Nicholas Love describes as "imagynatif deuotion" for a devout soul trying to bring themselves into contemplation.[5] However, Julian pushes her readers into a stronger understanding of how God has blessed her sight with an understanding of God's love. Christ asks Julian to climb into an inward understanding to how his love manifests in the passion.

At the beginning of her extended vision depicting the nature of sin, Julian presents her readers with a glimpse of the inner workings of God. This later vision, according to Julian, became fully realised to her "for twenty yere after the time of the shewing, save thre months" indicating that revelation from God extends beyond the moment of Julian's bodily seeing (Chapter 51, l. 73). Instead, Julian's study and prayer as an anchoress has allowed for God to shift her understanding to "seeing inwardly, with avisement" (51, ll. 76–77). The distance between Julian's revelatory experience and the advisement has led to her clarified vision of the formlessness of sin and God's love. The vision then gives us an incredible portrait of God in his full majesty. Julian slowly takes apart her vision in a way that exposes God's essential qualities to her "even-Christen". Despite the distance created by twenty years and the presumed distance of "seeing inwardly" as opposed to her bodily seeing, Julian is given access to a glimpse inside God. She is clear what she has witnessed is a part

of the totality that is hidden from men's understanding because "man is blinded in this life, and therefore we may not se oure fader" (Chapter 51, l. 120). Julian offers her inward seeing to give her "even-Christen" the substance of what is contained in what they cannot see. She offers guidance on how to interpret each feature her seeing has uncovered. We get a description of God that is both possible to imagine and contains the indescribable:

> "His chere was merciful. The colour of his face was fair brown, with full seemly countenance. His eyen were blake, most fair and seemly, shewing full of lovely pitte, and within him a hey ward, long and brode, all full of endless hevens" (51, ll. 105–108).

Julian's readers can both imagine and interpret the elements of God's appearance as she has illustrated them. She guides her readers through the meaning of each colour present in God's appearance so that there is no room for misinterpreting the surface. Then, Julian takes her readers within the substance of God. As Christ brought Julian to a view inside his side in her more conventional vision, Julian now brings her readers *within* her glimpse of God seen through years of contemplation. As in Christ's side, she and her readers there find the indescribable glory of all the heavens and, at the same time, a safe resting place, the "hey ward", where human souls can find continual rest.

Julian's decision to continue engaging with her visions allows for moments where she can strip away the blindness surrounding elements of the divine. What she reveals is at once comforting and impossible to fully grasp. Julian's work, through her "avisement" takes her readership through a performance most clearly recognised in Christian monasticism. Ineke van 't Spikjer's investigation of Hugh of St. Victor's writings reveals how the monastic author conceives of the human soul becoming aware of itself in seeking God. Hugh envisions how, in Spikjer's words, "the divine light proceeds from one level to the next towards man, man. . . has to return by the stages—from the visible to the invisible" (van t' Spikjer 2008, p. 81). There is a twofold way in which people can view the invisible nature of God: human reason inspired by God and revelation that is a direct gift of God's charity. Julian of Norwich gives her readers an insight into the inner workings of both. Her combined acts of bodily and inward seeing creates the space "between the visible and the invisible and the very point at which one turns into the other" (van t' Spikjer 2008, p. 84). Julian reveals her meditative work in the years following her revelations as a manner to instruct in Christian practices of meditation.

On a broader level, Julian is exposing a transcendent spiritual act. Her extended meditation on the master and the servant, created after her years of prayer, makes visible the inventive work of meditation. Mary Carruthers describes the fine-tuned practice of memory and imaginative work as "monastic rhetoric". Julian's engagement with finding the heart of her visions uses a combination of "verbal and visual media, their often synaesthetic literature and architecture. . . given a major impetus by the tools of monastic memory work" (Carruthers 1998, p. 3). However, unlike in traditional monastic practice, Julian's community is not confined to a set of religious practising prayer in ordinary measures around her. Her writing reflects the playing space she has created inside the intensity of her visions. As Denys Turner explains, Julian's theology within the text is "elicited through a process of progressive intensification and complex elaboration of particular and personal experience" (Turner 2011, p. xi). Instead, Julian focuses her revelatory work on her practice, creating a space where her readers could imagine themselves practising the substance of her meditations.

The community is essential to meditative practice. Mary Carruthers identifies how "the individual always had his or her being within a larger community, within which a single life was 'perfected', 'made complete,' by acquiring a civic being and identity" (Carruthers 1998, p. 2). Julian positions herself inside Norwich's spiritual life, at the very hub of communal religious practice in a major East Anglian city a generation out from a devastating encounter with the plague. Norwich Cathedral itself underwent a major building and remodelling project filled with images of both Christ's Passion and various apocalypse images in the years following Julian's revelations. Julian's anchoritic

encasement in the heart of Norwich becomes part of the documented spiritual history of Norwich. Christopher Abbot argues that Julian's physical presence carried more spiritual impact during her life in "some kind of local fame as a wise spiritual counsellor" than her revelations as recorded in either *Revelations of Divine Love* or her *Shewings* (Abbot 1999, p. 1). Julian is part of Norwich's spiritual landscape as much as the church that encloses her. Abbot stresses how Julian's text pulls the church around her visions through "an obligation to overcome that isolation without compromising her personal vision" (Abbot 1999, p. 61). Julian strips the imagery of her visions of what the Lollards view as a potential danger of "evil lurking within even the most sacred imagery, and that to the Lollards idolatry was perceived to have the dangerous potential to unleash the demonic" by her meditative practice (Sekules 2006, p. 301). She longs to have an understanding of her visions because she is "sterede in cherite to mine evenchristen that they might all see and know the same that I sawe" (Chapter 8, ll. 22–23).

Julian exposes herself as a revelatory figure by looking at, and into the love of God she has told us she yearned to the point of deadly sickness. Her longing for and bliss in the love of God has no end at the point of her mortal death but a perfection. As *The Cloud of Unknowing* author explains at the beginning of his text, a true devotional connection with God "may bi grace be bigonnen here, bot it schal ever laste with outen eende in the blis of heven" (TEAMS Middle English Text Series 1997, *The Cloud of Unknowing*. ll. 227–28). Julian passes through several miniature deaths in the creation of her book: her sensation of the lower half of her body dying, those in the room fearing she has already passed on, and her continuous state of death to the world as an anchoress. As a result, the continuous nature of monastic practice, where "the monks saw their contemplative lives pre-mortem as continuous with the next" is expanded to all Julian's readers through her practice (Turner 2011, p. 6). The resulting text places Julian at a nexus point where she is receiving and then conveying the love of God. She has become the place where God makes his love understood.

The closeness permitted by Christ in Julian's visions encapsulates several variations in the act of seeing. Julian of Norwich's central image of the lord and servant regarding God's love in the face of sin sees repentance as an act presented as "otherwise is the beholding of God, and otherwise is the beholding of man" (Chapter 52, l. 58). What Julian does is bring her readers an insight into what the "beholding of God" entails as far as his love for the sinful world. Through God's charity, Julian's readers can glimpse what it is to be seen by God in his compassion and love. Julian's strong use of colourful imagery and the immediacy of her visions broken down as Julian reports them in a way that mimics the revelatory experience of the apostle John crafts the bodily sight for the imagination of her readers that she received through her visions. Julian perfects her individual meditation through how her readers can experience the same form of seeing and being seen by the love of Christ she experienced and that it is a "comfort to them" (8, ll. 23–24). Julian's vision becomes a fully present image meant to be received by all believers. The longing soul, in seeking to pierce through the darkness to understand the divine, receives a fuller understanding that they are seen by the divine.

Julian's *A Revelation of Love* is meant to guide its readers to truth through interpretation and embodiment. It also serves as an example of the seeking modes of prayer Julian hopes to inspire in her readership. Julian has passed through a two-fold darkness, first in her physical illness and then in the doubt and consternation she describes in her earlier *Shewings* upon her first attempts to describe the gift she has received from her priest. She comforts her uncertain readers in the manner that God has demonstrated to her. She assures them that, though they may pray a "long time" without comfort, she has been made "seker by oure lordes mening that. . . He wille that we have knowing in himself that he is being" (Chapter 42, ll. 21–23). The 'gostly understanding' given to Julian is meant to help sinful creatures learn to constantly seek after God's love for them since "by the meekesse that we get in the sight of our sinne—faithfully knowing his everlasting love" (Chapter 82, l. 17). Far from the passive connotation of "knowing" that suggests knowledge received, Julian

describes knowing as an action to be undertaken "faithfully" by remembering human sin and shortcomings. Knowing becomes a two-fold act. Julian actively seeks out the action of longing to know God. God reveals his love to her in his visions, so there is no need for her to seek knowledge. Instead, she applies herself through faith to position herself inside a knowledge of the love of God. In this way, Julian's use of revelation is actually closer to Paul's prayer for the church in Rome: that they might be faithful because of God's good news demonstrated through "the revelation of the mystery hidden for long ages past, but now revealed and made known through the prophetic writings by the command of the eternal God" (Romans 16:25b–26a). Julian's visionary experience compels her to make plain the truth of God's love.

Within the context of *A Revelation of Love*, Julian believes God shows—that is, reveals—to her both through her reason and through her imagination. Julian presents those revelations that can be described as spiritual manifestations in such a way that suggests she has spent time evaluating what these visions signify and how best to present them to her readers. Her vivid imagery, as well as her closeness to the figure of the dying Christ participates in a trend in medieval English vernacular verse that Christopher Abbot argues centres on "a Christ whose death is presented as an act of personal love for every individual" (Abbot 1999, p. 52). When describing the sight of the bleeding Christ on the cross, Julian's imagery becomes visceral: 'gret droppes of blode fellle downe fro under the garlonde [the crown of thorns] like pelottes. . .and in the coming out they were browne rede, for the blode was full thicke' (Chapter 7, ll. 10–12). Her vision of Christ proceeds from an image of the cross brought into Julian while she is lying on what she presumes to be her deathbed by a curate. The priest urges Julian to "looke thereupon and comfort thee therewith" (Chapter 3, ll. 18–19). God gifts Julian with the intense images. More importantly, God gives Julian the will to fashion her revelations into a work of theological significance. Julian reports how she is given "space and time to beholde" the truth of God's love for the world (3, l. 19). This continues from the graphic imagery present in Julian's sight as shared with her readers, into a spiritual or "gostely" sight which "dwelled in [her] understonding" (3, l. 20). Julian uses extreme and colourful images to invoke in her readers the same sense of piety and divine feeling she had while on her sickbed, then guides her readers towards the meaning God gifted to her through her later meditations. *Revelations of Love* provides Julian's Christian community with eyes through which they can participate in a vivid performance of Christ's love for them.

## 5. Conclusions

Part of Julian's work in performing her revelations to her "even-Christen" is the prayerful work of memory. Julian works through her visions in the way the monastic thinker is made to "memorise" the scripture. We sit as fellow spectators witnessing Julian marvel over "the dark, difficult parts in the text which one can isolate, holding one or two up at a time to the audience for admiration, and proceeding to loosen up its knots and expand its meaning as one explicates and expounds" (Carruthers 1998, p. 65). In the context of the extreme pressure brought onto all manner of preaching and teaching by the crackdown on the Lollard heresy, Julian's rhetoric surrounding the memory of her visions is, by necessity, a careful balancing act. Her work in the intervening years is not on her own study but the longing for continued sight, for a seeing with her spiritual eye what God has intended "in general" or for all his people. Her language towards her revelations is firmly turned toward the communal body of believers, and she claims that nothing she has seen is a secret God has not shared with the whole community. Barry Windeatt argues that a transition occurs in Julian's writings between *A Revelation of Love* and a shorter, earlier work, Shewings, which contains less detailed depictions of her visions. He seems almost surprised that the incredible scenes Julian's words create "reflect the serene, assured outcomes of intervening meditation" (Windeatt 2004, p. 71). Julian's trust in God's interpretation of his own visions gives her the confidence to position what she

does as a practice merely in seeing and sharing even as she uncovers the substance of her visions.

Julian's collection of showing must be referred to as a revelation to grasp the spiritual scope this work hopes to achieve. By positioning herself as both a performer and fellow spectator of Christ's revelations to those he loves, Julian transforms herself into a body constantly seeking God's message of love for his people. She guides her readers through her desire and prayer so that they can journey with her from the revelation of her visions into an understanding of the simple message God has contained there. God reveals his love to Julian in the context of her work, not only through intense spiritual inspiration but also through providing her with the scripture and intellect required to interpret and present her showings. She, in turn, reveals the love of God through her text: acting as a conduit, interpreter and secondary gift giver of the truth her work contains. *A Revelation of Love* demonstrates Julian of Norwich's life-long effort to make sense of and impart to her fellow believers the devastating simplicity of God's grace and charity.

**Funding:** This research received no external funding.

**Data Availability Statement:** All research is available through critical editions of Middle English texts.

**Conflicts of Interest:** The author declares no conflict of interest.

## Notes

[1]  All in-text citations from *Revelation of Divine Love* are taken from the critical edition by Nicholas Watson and Jacqueline Jenkins. (Julian of Norwich 2006).

[2]  Julian uses this phrase throughout her works to describe the community of all Christian believers.

[3]  Note Julian's struggling and striving after light at the beginning of her Second Revelation and God's reminder to her that "if God will shew thee more, he shal be thy light" (Chapter X 9–10).

[4]  'We love because he first loved us' 1 John 4:19. All Biblical references come from the English Standard Version.

[5]  See Nicholas Love's *Mirror of the Blessed Love of Christ* for a full description of how the English conceptualised the move from active to contemplative religious practice through the practice of affective piety. Importantly, the bodily image of hearts heads Love's treatment, but focuses instead on the model set by St. Cecila, who "bare alwey þe gospel of criste hidde in her breste. . . she chace certayne parties most deuoute" (Love 11). In Love's reading, Cecila's incorporation of the text seems notably visceral.

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
