# Peer review of "Mystic on a Tilting Stage: Julian of Norwich’s Performance of English Visionary Devotion"

_religions, doi:10.3390/rel14121466_

Round 1

Reviewer 1 Report

Comments and Suggestions for Authors

The main aspect lacking in this article is framing. The author needs to explain in more intentional ways why this study is important and original. The article comes across as vague and disorganized, with important points (e.g., the influence of the Lollards, the aesthetic setting of Norwich Cathedral) buried in the middle of incoherent paragraphs. Some key points do get across, but the reader is left wondering why knowing this about Julian is essential. The paper needs more support from scholarly literature. For example, the author notes that a certain affect was typical of the fourteenth century but provides no footnote to reference studies on this theme. The author should cite a critical edition or manuscript of Julian’s works rather than provide mere chapter references in parentheticals. The comparison with Hinduism starting on line 265 seems to be little more than free association and should probably be cut.

Comments on the Quality of English Language

The English in the paper needs a significant overhaul. Multiple examples of sentence fragments, run-on sentences, shifts in verb tenses, redundancy of words and phrases, etc. Too many sentences and even a few entire paragraphs are awkward and/or unclear.

Author Response

Thank you for your helpful feedback. I have used the performative nature of Julian's interaction with her visions as a new frame for the text. The emphasis should now center on how Julian's revelations address the concerns of her devotional community and how she creates a space for her reader's spiritual performance. Work with Jennifer Summit on the transcendent nature of the Cressey edition, and an early reference to Nicholas Love help to strengthen this position. Although I initially brought more research into my exploration of darsan in Julian's work, I have chosen for this version to cut the references for brevity and clarity. Thank you again.

Reviewer 2 Report

Comments and Suggestions for Authors

This article explores Julian of Norwich’s Revelation of Love as a public-facing work of revelatory knowledge, directed toward the theological enlightenment of the Christian community. The author’s unique contribution seems to be in connecting Julian’s mystical revelations in the late Middle Ages to a biblical model of revelation established by John of Patmos in the book of Revelation. In my view, the author is only partially successful in demonstrating a connection (in form and content) between Julian’s revelatory experiences and those described in the New Testament text. Further revisions to the argument are necessary to demonstrate the utility of painting Julian as a literary visionary in the style and tradition of specifically John of Patmos, particularly in how this connection advances scholarship on Julian, vernacular mysticism and public devotion of the late Middle Ages.

On page 1, line 35, the author asserts (but does not prove or demonstrate) that “Julian’s most obvious literary predecessor is the Apostle John.” First, in my understanding of the biblical scholarship, it remains contested within the field whether John of Patmos (author of Revelation) is identical with the apostle John (and indeed further, whether the Gospel of John is authored by an apostle who knew Jesus during his lifetime). Second, it is unclear to my why the author singles out Revelation as a biblical text uniquely relevant to Julian’s literary work when other visionary/apocalyptic texts exist in the Bible. While Revelation is unique to the New Testament, the books of Isaiah, Ezekiel, and Daniel offer additional biblical accounts of apocalyptic literature where a visionary is given knowledge of/by God through means of bodily sight. Is Revelation distinct amongst biblical visionary texts in a way that makes it uniquely appropriate to understanding and interpreting Julian? The author has not demonstrated this, nor does he/she appear to have even considered the question. If the author is drawing on a larger body of secondary or biblical scholarship, further citations are needed here to flesh out this connection for his/her readers. If this is an insight unique to the author, he/she needs to develop this argument much further, to demonstrate how or whether Revelation is distinct among biblical visionary literature, such that it and it alone provides a helpful lens for understanding Julian.

There are other ways that the author’s reading of Julian seems idiosyncratic, or at least unsupported by the larger field of Julian scholarship. For example, on pg 1 (lines 28-31) the author posits that Julian’s self-understanding is not that of a theologian, but rather of “a working body and spirit in motion, transmitting back the simple truth of God’s love.” Theologians self-evidently also are bodies and spirits in motion, whose work (among many tasks) involves transmitting the truth of God’s love through their writing. While as a Medieval woman, Julian needs to be diplomatic about her language, and the authority she claimed for herself (particularly when expounding on topics, such as universal salvation, that could be interpreted as contradicting church teaching), recent scholarship on Julian has taken pains precisely to acknowledge her writing as an explicit work of theology. Denys Turner’s book Julian of Norwich: Theologian (Yale, 2011) makes this precise point. The author of this article repeatedly acknowledges (for example, on pg 6, lines 230-231) that Julian required time to reflect on the meaning of her initial visions, in order to draw conclusions for her community. While it is certainly possible to describe such reflection as supported by the Holy Spirit in Julian’s prayer and meditation, reflecting on the meaning of an insight about God/the world is precisely the work of the theologian. It seems to me that the author is setting up a false dichotomy between Julian as a “theologian” and Julian as a devotional visionary, who nonetheless offers her visions to a public community for their benefit and the building up of the Kingdom. I am unclear what the utility is in interpreting Julian in this way, which seems to reject the possibility of taking her seriously as a deep and orthodox theologian.

The author’s strengths in this piece lie in making original and interesting connections between Julian’s work and other fields and genres of scholarship not normally associated with Julian. However, the author is not able in this current draft to provide the scholarly scaffolding to undergird these novel connections and demonstrate their relevance to more “mainstream” scholarship about Julian.

There is an instructive example of this phenomenon in the paragraph that runs between pages 6 and 7 (lines 265-290). In this paragraph, the author references the Hindu practice of “darshan” to explain the way Julian’s prose creates an imaginative framework within which her readers can “behold” God. The author’s consideration of darshan, a theologically rich devotional practice central to Hinduism, is limited to a description of a photography exhibition (presumably viewed by the author) hosted by the Birmingham Museum of Art. While this exhibit might be the author’s first introduction to the concept of “darshan,” there is a wealth of scholarly writing on darshan that the author fails to consult or cite in explaining this concept. If the author chooses to retain this inter-religious comparison, I suggest that he/she consult Diana Eck’s excellent book Darsan: Seeing the Divine Image in India (Columbia, 1998). In my limited understanding of darshan, key to the way this practice works is the presence of a physical object within or through which the divine can communicate/access people. While Julian might want to convey a concrete picture to her readers through her words, she certainly isn’t constructing a physical object which, after being ritually blessed, will allow worshippers direct access to the divine.

To reiterate, the author’s strength in this piece is connecting scholarship on Julian to other genres and topics not typically associated with Julian – most significantly the biblical text of Revelation. The only scholarly support I could find for this claim, within the paper, is a description (on pg. 3, lines 115-119) that the Norwich Cathedral (within which Julian was anchored) is decorated with biblical images from John’s apocalypse. While I think there is a supportable argument that could be made about the effect a visionary’s environment might have on her subsequent theological reflections and understanding, I don’t think this architectural detail can provide sufficient support to the claim that John’s literary account, as presented in the New Testament, is truly Julian’s most significant literary predecessor. The claim that the biblical text of Revelation uniquely illuminates Julian’s own visionary literature seems insufficiently supported, if this is the only support the author has from secondary or biblical literature.

Comments on the Quality of English Language

Overall, the English prose appears error free, but there are a few places that require further copy editing. The author uses the Middle English term "evencristen" throughout the paper will little contextual clues to the reader that this is a technical term employed by Julian. The first time this term appears (line 34 of pg 1), the author used the term without quotation marks or an internal citation to Julian's text. Since Julian's work appeals to a wide range of scholars (from modern systematic theologians to, theoretically, biblical scholars who might be interested in the author's argument connecting Julian's work to the biblical book of Revelation), it would be helpful to provide a definition or translation of this terms, at least in the first instance it is used in the midst of otherwise contemporary English prose.

There are a few instances where the author's sentence construction is difficult to follow. I had a particularly hard time locating subject/object agreement in the sentence on pg. 3 (lines 103-105) beginning with "Julian imparting...". 

On pg 6, in the sentence immediately following footnote 17 (lines 252-254) "pointed" should be "pointing." 

Author Response

Thank you so much for your feedback. I hope this new version addresses your concerns about the disjointed connections that troubled in the first version. I have reframed the article to focus on Julian as "performing" her work as a revelator. This shifts her relationship with John as an exploration of how the two position themselves as receivers and interpreters of God's presence in their work. With this new framework, I believe the section on vision is much strengthened. The sections on Norwich itself become more of an exploration of the staging to which Julian aligns her descriptions of her visions. I have moved my work with Nicholas Love to the beginning of the article to emphasize how Julian's writing works as a powerful culmination of imaginative performance common to English devotional practice. Thank you for your recommendation of Diane Eck's book. I eventually cut the new section on darsan for the ake of brevity but found it helpful to fine-tune my understanding of the gift of vision. I have read my work over for wordiness and difficult phrasing. Thank you again for your thoughtful comments. 

Reviewer 3 Report

Comments and Suggestions for Authors

Please see the file below.

Comments on the Quality of English Language

See attached file

Author Response

Thank you so much for your thoughtful feedback. I hope that this version helps to give more context to the close reading by focusing on how Julian uses her visions as a performance of her community's devotional practice. I have moved my exploration of Julian's connection to Nicholas Love's Mirror to the beginning of the essay and proceeded to explore how her work leads her to multiple understandings of her participation in Christ's love. This, I hope, pushes your interest in how I explore Julian's relation of knowing and action to the forefront. My exploration of John now follows this performance framework, focusing on how the two authors position their bodies as authors and receivers of divine mystery. I have parsed down some of my thornier sentence structure, and I hope this version is clearer. Thank you again for taking the time with my work.

Round 2

Reviewer 1 Report

Comments and Suggestions for Authors

Much improved over the previous draft. Well done.

Author Response

Thank you so much.

Reviewer 3 Report

Comments and Suggestions for Authors

The author appears to have responded in good faith to the reviewers’ comments, and the essay as it stands presents a broadly elucidating perspective on Julian’s revelatory text. However, it remains ultimately constrained by the lack of argumentative thrust (this is not helped by the poetic, but rather vague title: a clearer and more directed title could provide a better hook). Is this an essay about devotion as performance? If so, what is the original argument about Julian’s text? Or, is it about Julian in scriptural and devotional context? As it stands, the idea that Julian participates in a performative devotional milieu is well-trodden ground in the field, and despite the comparison with Nicholas Love and John of the Apocalypse, this essay has not formed a sufficiently coherent argument to call it an original contribution.

Syntax and technical language are still deployed in slightly confusing ways, often without identifying a clear operation, e.g. 42-43: ‘The performative nature of imaginative devotion as described in English literature is revealed in the instructions given to the soul seeking God.’  Another instance: language of ‘intention’ is engaged at various points (79, 95, 111) without explaining how or why we can presume to know Julian’s intentions, or any reference to the well-established medieval contemplative concept of intentionality.

Given that these fundamental issues – of argumentative coherence, direction, and language – have not been sufficiently addressed, I am afraid I cannot recommend publication at this time.

Comments on the Quality of English Language

See above.

Author Response

Thank you so much for the second read through.